# Response Surface Optimization of Solar Drying Conditions and the Effect on the Quality Attributes and Drying Characteristics of Qula Casein

**DOI:** 10.3390/foods11162406

**Published:** 2022-08-10

**Authors:** Jun Wang, Lina Wang, Linlin Wang, Ling Han, Lianhong Chen, Shanhu Tang, Pengcheng Wen

**Affiliations:** 1College of Food Science and Technology, Southwest Minzu University, Chengdu 610041, China; 2College of Food Science and Engineering, Northwest A&F University, Xianyang 712100, China; 3College of Food Science and Engineering, Gansu Agricultural University, Lanzhou 730070, China

**Keywords:** Qula, solar drying, drying characteristics, response surface methodology, lipid oxidation

## Abstract

The objective of this study was to investigate the potential application of a mixed-mode solar dryer to Qula dehydration in the Qinghai-Tibet Plateau of China. A three-factor five-level central composite rotatable design (CCD) of the response surface methodology (RSM) was employed to evaluate the influences of drying process variables on quality attributes in terms of lipid oxidation [peroxide (POV) and thiobarbituric acid reactive substance (TBARS)] and Maillard reaction (5-hydroxymethylfurfural, 5-HMF). The impact of drying temperature (30–50 °C), material thickness (5–15 mm), and wind velocity (0.4–1.4 m/s) on the color, POV, TBARS, and 5-HMF of Qula were studied. Optimum operating conditions were found to be a temperature of 43.0 °C, material thickness of 11.0 mm, and wind velocity of 1.0 m/s based on the minimum of POV, TBARS, and 5-HMF. In this condition, the values of POV, TBARS, and 5-HMF were 0.65 meq/kg, 0.516 mg/kg, and 4.586 mg water/L, respectively, which are significantly lower than for drying of Qula by open sun-drying (*p* < 0.05). Compared with open sun-drying, the drying time of Qula by solar drying was shortened by 61.5%. The results indicate that the mixed-mode solar dryer is a feasibility drying technology for Qula and could effectively improve the quality of products.

## 1. Introduction

Qula (Tibetan name), made from yak milk by defatting, acidifying, and drying, is one of the most important indigenous products in the Qinghai-Tibet Plateau, China. Qula is an indispensable food for herdsmen in their daily lives, which is rich in protein, lactose, and polyunsaturated fatty acids [1,2]. It is also the main raw material for yak milk casein products. In China, more than 90% of casein is purified from Qula, with a great market demand [2]. Due to the rich essential amino acid and trace element content, yak milk casein is widely used in the food industry; it is also essential in the pharmaceutical, cosmetic, coating, and leather industries [3,4].

However, the quality attributes and functional properties of casein are remarkably affected by the quality of Qula, especially the color. Several attempts have been made to improve the casein quality by dissolution, discoloration, and re-drying of Qula. Liu et al. [2] found that the functional and structural properties of yak caseins are significantly affected by pH. Various enzymes, including angiotensin I-converting enzyme, dipeptidyl carboxypeptidase, and α-chymotrypsin, are also used to improve the quality and utilization of yak casein [3,5]. However, such approaches have failed to address the fundamental issue of the raw material itself. Therefore, the most important thing is to solve the quality problem of the Qula raw material. Thus, the quality of Qula, especially its color, should be improved to enhance the quality of casein products. 

Thus far, Qula is still made by traditional methods, including defatting (by a cream separator), acidification (natural fermentation), and drying (natural open sun-drying), in China [2,5]. Among these methods, drying plays a critical role in the maintenance of quality, including color deterioration, fat oxidation, Maillard reaction, and microbial contamination. Drying is a process that prevents the growth and reproduction of microorganisms; increases shelf life; transforms the food into a new product; and reduces packaging, storage, and transportation by reducing water activity to adequate levels, thus avoiding microbial deterioration and quality losses due to biochemical reactions [6]. However, over the years, in Qinghai-Tibet Plateau, China, open sun-drying is still being used for Qula dehydration, which strongly depends on weather conditions, exposure to environmental contamination (dust, insects, and birds), and mycotoxin contamination, resulting in a slow drying rate and a poor quality of the products [7]. Therefore, a mechanical drying technology and useful drying equipment should be explored for Qula products.

Remarkable energy consumption is the main challenge faced by mechanical drying. Herein, solar energy is of interest in the whole world because it is clean, environmentally friendly, and inexhaustible. The northwestern part of China has abundant solar resources, and the Qinghai-Tibet Plateau has the most abundant solar resource in China. The area experiences more than 2200 h of sunshine per year and radiation of more than 250 W/m^2^ in April to August, which is the main period of Qula production [8,9]. Therefore, a mixed-mode solar dryer was developed in our previous work, and the potential impact and application value of this new solar dryer should be studied and expounded in Qula drying based on the quality attributes.

Color, which extremely influences the subsequent processing, sensory attractiveness, quality, and final price of Qula, is the most intuitive and important quality attribute of Qula. Color is influenced by many factors, including drying temperature and time of drying, oxidation, enzymatic and nonenzymatic browning, moisture content, and surface moisture [10]. Lipid oxidation and the Maillard reaction contribute considerably to the formation of an undesirable color of Qula [2]. Lipid oxidation has received much attention because of its undesirable implications for human health and its contribution to a decrease in the nutritional value, flavor compounds, and color quality of dairy foods [11]. Lipid oxidation is also the main cause of the deterioration of lipids and lipid-containing foodstuffs, which can occur in dairy products due to the imbalance of pro-oxidants and antioxidants [12]. POV and TBARS have been used to evaluate the level of lipid oxidation [13]. Nonenzymatic browning, known as the Maillard reaction, is one of the major detrimental reactions that cause the formation of chemically stable and nutritionally unavailable derivatives of dairy products [14]. Nonenzymatic browning is a complex chain of reactions that is initiated between an amine group on a protein/peptide and a reducing sugar [15]. 5-HMF is a complex compound that forms as an intermediate in the Maillard reaction and the caramelization of many sugary foods during heat treatment [16,17] and is frequently used to assess the extent of the Maillard reaction. Many factors in the food processing environment, such as pH, pressure, moisture, temperature, and food composition, affect the production and reaction process of 5-HMF [18]. 

Therefore, the main aims of this study were to evaluate the potential application of solar drying in Qula drying based on quality changes, including POV, TBARS, and 5-HMF. A three-factor five-level CCD of RSM was employed to study the impact of drying process variables (drying temperature, materials thickness, and hot air velocity (wind velocity)) on the quality attributes. The moisture diffusion rate and quality attributes were compared of Qula between optimized solar drying and open sun-drying.

## 2. Materials and Methods

### 2.1. Material Preparation

Yak milk was obtained from a yak breeding demonstration base in the Gannan pastoral areas of Gansu Province, China. After removing debris, such as hair and dust, yak milk was defatted using a cream separator and fermented for 48 h at a temperature of 20–25 °C (Figure 1) [19]. Fermented and completely solidified casein was drained to remove whey and dehydrated using the solar dryer employed in the current work, which is described in Section 2.2. The whey content of Qula was 52.6 ± 0.68% (wet base) prior to drying, which was determined in accordance with the method reported in previous research [2]. The raw Qula casein is a milky white granular material with an average moisture content of about 62.50%, fat content of 2.15 ± 0.26%, protein content of 32.43 ± 0.75%, and lactose content of 1.69 ± 0.18%.

### 2.2. Mixed-Mode Solar Dryer

This solar dryer was fabricated in the region of Qinghai-Tibet Plateau, China, located in the pastoral area in Xiahe county (101°54′–103°25′ N latitude, 34°32′–35°34′ E longitude, and an altitude of 3000 m), which receives solar radiation of more than 2200 h per year [8]. A schematic diagram and picture of the facility employed in the present study are illustrated in Figure 2A and Figure 2B, respectively. The mixed-mode solar dryer mainly consists of a solar collector system and drying chamber exhaust system (chimney). The dimensions of the drying chamber are 1.2 m long, 1.0 m in height, and 0.8 m in width and of the solar collector are 2 m long and 1 m in width. Five trays with equally spacing were arranged in the drying chamber. The temperature was measured by a temperature sensor with an accuracy of ±0.3 °C (PT100, TD Sensors Technology, Shanghai, China), which was placed at the inlet and outlet of the drying chamber. The wind velocity was detected by a digital anemometer (AS8836, Smart Sensor, Shenyang, China, with an accuracy of ±0.001 m/s) placed at the hot air inlet and outlet. After the test began, the tray was taken out every 1 h, weighed, and recorded until the dry base moisture content of the material was lower than 12.0%, which was the drying end point, and the test was stopped.

### 2.3. Experimental Design

The RSM is a representative statistical method that has been successfully used to develop, improve, and optimize processes, as well as being widely used to optimize conditions in agricultural and engineering research. In our study, the RSM was employed to study the effects of the drying parameters, i.e., temperature (*X*_1_), material thickness (*X*_2_), and wind velocity (*X*_3_), on the quality attributes (*Y*_1_–*Y*_3_, namely POV, TBARS values, and 5-HMF contents, respectively). In accordance with our preliminary study, the ranges selected for the temperature (30–50 °C), material thickness (5.0–15.0 mm), and wind velocity (0.4–1.4 m/s) are presented in Table 1. A 3-factor 5-level CCD was used to evaluate the major and combined influences of drying conditions on the quality of Qula, and mathematical models were established between the variables (drying conditions) and the response variables (POV, TBARS, and 5-HMF). Twenty treatments were assigned in accordance with the second-order CCD with three independent variables (Table 2). The experiments were randomized to minimize the impacts of unexplained variability in actual responses. In CCD, center points were repeated six times to calculate the repeatability of the method. As shown in Table 2, the CCD matrix included the values corresponding to the levels of factors and treatments and allowed the development of the appropriate empirical equations [20]. On the basis of the optimized solar drying condition, drying kinetics and quality attributes (POV, TBARS, 5-HMF, and color) were further analyzed. Open sun-drying was used as the control group.

### 2.4. Quality Attributes’ Determination

#### 2.4.1. POV

The POV of Qula was performed according to the method of Sun et al. [21]. The crushed Qula sample (2.0 g) was treated with a 30 mL of chloroform and acetic acid mixture with a ratio of 2:3. The mixture was shaken vigorously at room temperature for 20 min. Subsequently, 0.5 mL of saturated potassium iodide (KI) solution was added into the mixture. The mixture was kept in darkness for 5 min, followed by the addition of 75 mL of distilled water and mixed. Then, 0.5 mL of starch solution (1%, *w*/*v*) was added into the mixture as an indicator. The POV was obtained by titrating the iodine liberated from the KI solution with a 0.01 M standard sodium thiosulfate solution. The POV is expressed as milliequivalents peroxide/kg in Qula.

#### 2.4.2. TBARS

TBARS was determined according to the method described by Sun et al. [21], with a slight modification. The crushed Qula sample (2.0 g) was dissolved completely with an alkaline solution (pH 8.0) at 65 °C [2] and mixed with the thiobarbituric acid solution (1.0%, *v*/*v*) and trichloroacetic acid (5.0%, *w*/*v*) in a 1:1 (*v*/*v*) ratio. The mixture was shaken vigorously at 90 °C for 45 min. After cooling, the mixture was centrifuged at 3000 rpm for 5 min, and the absorbency was measured immediately at 532 nm with a spectrophotometer (UV2550, Shimadzu Corporation, Kyoto, Japan). The TBARS content is expressed as mg MDA/kg in Qula.

#### 2.4.3. 5-HMF

The 5-HMF contents of Qula were determined according to the modified method of Gao et al. [22]. The crushed Qula sample (2.0 g) was treated with 10 mL of 0.15 M oxalic acid solution. Then, 3 mL of potassium hexacyanoferrate solution (90 g/L) and 3 mL of zinc acetate solution (183 g/L) were added. The mixture was then shaken vigorously at room temperature for 10 min. The total volume of the mixture was kept to 50 mL with acetonitrile. Then, the centrifugation procedure was carried out at 5000 rpm for 10 min. Subsequently, the supernatant was filtered with a 0.45 μm syringe filter and injected into the high-performance liquid chromatography (HPLC) system (Agilent 1100, Agilent Technologies Co. LTD, Santa Clara, CA, USA) equipped with a C18 column. The mobile phase, flow rate, and injection volume were the solvent mixture (methanol: ultrapure water, 15:85), 1.0 mL/min, and 20 μL, respectively. The wavelength of the DAD detector was 280 nm. The 5-HMF concentrations of the molasses samples were determined by the calibration curve of the 5-HMF. 

#### 2.4.4. Nutritional Components and Color Parameters

According to the methods reported by Liu et al. [2], the moisture, protein, fat, and ash content of Qula from solar drying and open sun-drying were measured. The color parameters in terms of *L*^*^ (whiteness or brightness), *a^*^* (redness/greenness), and *b^*^* (yellowness/blueness) were also detected by a CR-10 Colorimeter (Shanghai konica minolta investment Co., Ltd., Shanghai, China).

### 2.5. Drying Kinetics

#### 2.5.1. Drying Curves

The moisture ratio (*MR*) of the Qula during drying was determined according to Equation (1) [23],
(1)$$MR=\frac{M_{t}-M_{e}}{M_{0}-M_{e}}$$
where *M*_0_, *M_e_*, and *M_t_* represent the initial moisture content, equilibrium moisture content, and moisture content at *t* time of drying (kg water/kg d. b), respectively. All moisture content is expressed on a dry basis (d. b).

#### 2.5.2. Mathematical Modeling

To model the Qula dehydration process, seven thin-layer equations (Table 3) were employed [23,24]. The coefficient of determination (*R*^2^), residual sum of squares (*RSS*), and reduced chi-squared parameter (*χ*^2^) between the predicted and experimental values were used to determine the quality of fit of the experimental data, which were computed by Equations (2)–(4):(2)$$\chi^{2}=\frac{\sum_{i=1}^{N}{\left(MR_{\exp,i}-MR_{pre,i}\right)}^{2}}{N-z}$$
(3)$$RSS=\sum_{i=1}^{N}{\left(MR_{pre,i}-MR_{\exp,i}\right)}^{2}$$
(4)$$R^{2}=\frac{\sum_{i=1}^{N}{\left(MR_{pre,i}-MR_{\exp,i}\right)}^{2}}{\sum_{i=1}^{N}{\left(\bar{MR_{pre}}-MR_{\exp,i}\right)}^{2}}$$
where *MR*_exp,*i*_ and *MR_pre_*_,*i*_ are the experimental and calculated dimensionless moisture ratios, respectively; *N* is the number of experiments; *z* is the number of constants.

### 2.6. Statistical Analysis

For the statistical analysis of the experiment, Design-Expert v.8.0.6 Trial (Stat-Ease, Inc., Minneapolis, MN, USA) was used. Meanwhile, the analysis of variance (ANOVA) was conducted to establish the optimal combination of experimental conditions for the solar drying procedure and investigate the potential relationship between the three responses and the experimental variables determined. The generalized response surface model for describing the variation in the response variables is as Equation (5),
*Y* = *β*_0_ + *β*_1_*X*_1_ + *β*_2_*X*_2_ + *β*_3_*X*_3_ + *β*_11_*X*_1_^2^ + *β*_22_*X*_2_^2^ + *β*_3*3*_*X*_3_^2^ + *β*_12_*X*_1_*X*_2_ + *β*_13_*X*_1_*X*_3_ + *β*_23_*X*_2_*X*_3_(5)
where *Y* is the estimated response; *β*_0_ is the model intercept; *β*_1_, *β*_2_, *β*_3_, *β*_11_, *β*_22_, *β*_33_, and *β*_12_, *β*_13_, *β*_23_ are the linear quadratic and interaction coefficients. The values of the coefficient of determination (*R*^2^), adjusted coefficient of determination (*R*^2^-adj), and predicted coefficient of determination (*R*^2^-pred) were used to judge the fitness of the model. *R*^2^ was defined as the ratio of the explained variation to the total variation; the proximity to unity of this ratio illustrated the model accuracy [32]. The high values of the adjusted and predicted *R*^2^ also indicated the high fitness of the model. Furthermore, the insignificant lack-of-fit further verified the model [33]. The correctness and fitness of an established quadratic model after the most accurate model was selected by ANOVA. The interaction of factors was observed through the surface and contour plots from the model. Adeq precision was used to measure the signal-to-noise ratio. A ratio higher than four is desirable, and the model can be used to navigate the design space [34,35].

## 3. Results and Discussion

### 3.1. Fitting Response Surface Models

The relationship between the POV, TBARS values, and 5-HMF contents and the three operating parameters (drying temperature, material thickness, and wind velocity) were fit to the following second-order polynomial Equations (6)–(8), respectively. *X*_1_, *X*_2_, and *X*_3_ are the coded values of drying temperature, material thickness, and wind velocity, respectively.
POV = 0.69 − 0.27*X*_1_ − 0.11*X*_2_ − 0.13*X*_3_ + 0.38*X*_1_^2^ + 0.33*X*_2_^2^ + 0.21*X*_3_^2^ − 0.08*X*_1_*X*_3_(6)
TBARS = 0.85 − 0.71X_1_ − 0.20*X*_2_ − 0.097*X*_3_ + 0.63*X*_1_^2^ + 0.30*X*_2_^2^ + 0.12*X*_3_^2^ +0.057*X*_1_*X*_2_ − 0.28*X*_1_*X*_3_ − 0.24*X*_2_*X*_3_
(7)
5-HMF contents = 7.16 − 4.26*X*_1_ − 3.16*X*_2_ − 2.09*X*_3_ + 2.44*X*_1_^2^ + 1.85*X*_2_^2^ + 1.81*X*_3_^2^ + 1.92*X*_1_*X*_2_ − 0.48*X*_1_*X*_3_ + 0.48*X*_2_*X*_3_(8)

The ANOVA of linear, quadratic, and interaction terms for each response variable and coefficient of the prediction models is presented in Table 4. The significance of each coefficient was determined by the *p*-value. A small *p*-value indicates a high contribution of the corresponding model term to the response variable [35,36]. Among the three coating parameters, temperature was the most influential parameter and wind velocity was the least influential parameter. Linear and quadratic coefficients were all significant (*p* < 0.01). Among the interactive coefficients, only the interaction effect of temperature and wind velocity for POV, the interaction effects between temperature and wind velocity and between material thickness and wind velocity for TBARS values, and the interaction effect of temperature and material thickness for 5-HMF content were significant (*p* < 0.01). The values of *R*^2^ > 0.96, *R*^2^-adj > 0.9, *R*^2^-pred > 0.70, adeq-precision > 4, and the maximum C.V of 13.10 < 14 (coefficient of variation) revealed that the data were fit well by the proposed polynomial regression model [37,38]. Furthermore, the fitness of the model was investigated using the lack-of-fit test (*p* > 0.05), which indicated the suitability of the models to predict the variation accurately [39].

### 3.2. Effects of Solar Drying Conditions on the POV

By using hot air drying, the unregulated drying air temperature and air velocity may impair the crust structure, which ultimately will lead to an undesirable color, texture, and structure of the dried product [20]. It is well known that the POV is a vital evaluation index to estimate the degree of lipid peroxidation. The POV is conducted to have information on the oxidation status of foods. A food system having a lower POV is usually perceived to have reasonable oxidative stability and shelf-life [12]. In our present research, the POV varied from 0.61 meq/kg to 2.15 meq/kg at different solar drying conditions. In the experiments, the lowest POV was obtained at a temperature of 40 °C, material thickness of 10 mm, and wind velocity of 0.9 m/s. The effects of three independent variables on the POV were determined by the coefficient of the second-order polynomial regression equations (Table 4). By neglecting nonsignificant terms, the fit model of the POV in the coded levels was constructed using Equation (6).

According to Equation (6), surface plots representing the relationship between process variables and the POV are shown in Figure 3A. The effects of temperature, material thickness, and wind velocity on the POV that had no significant difference are not shown in the present study (*p* > 0.05). The present results demonstrated the response surface and contour plots for the POV at constant drying material thickness (10 mm). A significant difference was observed between the interaction of temperature and wind velocity on the POV (Table 4, *p* < 0.05). The POV decreased with increasing temperature, but increased after 40 °C, which might be because increased peroxide was produced at a high drying temperature. This phenomenon led to a high POV. When drying to the same moisture content, increased time was consumed at the low-temperature range compared to that at a high temperature. Table 2 shows that the POV was smaller at high wind velocity than that at low wind velocity for the same temperature and material thickness conditions, which indicated that wind velocity plays a significant role in the lipid oxidation [40]. The main reason was that the level of the wind speed may affect the oxidation of fat by affecting the transfer and flow of water and the maintenance of temperature. The present results demonstrated that the obtained POV was relatively smaller at 40 °C than at other temperature points. This result is inconsistent with a previous report [41], which demonstrated that temperatures of 22–31 °C can easily lead to the further oxidation of hydroperoxides into secondary oxidation products, thereby causing the decline of the POV. The rise in the POV is due to the breakdown of fatty acids into oxidation products and analysis intervals, showing that an increased POV also shows pronounced changes in the fatty acids’ profile [42]. Furthermore, different POV results may be due to variations in the process conditions for different products and methods for the POV assay used [43]. Hydroperoxides are primary lipid oxidation products; their contents are predominantly determined by the ratio of formation to degradation, and the high-temperature process can lead to the lipid oxidation as the POV increases [41]. Some lipid oxidation products can be formed and affect the quality and flavor of Qula.

### 3.3. Effects of Solar Drying Conditions on the TBARS

Milk and dairy products contain about 23–25% unsaturated fatty acids, which are susceptible to auto-oxidation. In addition to the POV, TBARS values are frequently used to measure the formation of secondary oxidation products in lipid oxidation and are closely related to the sensory quality of dairy products. In the present study, TBARS values were changed from 0.519 mg/kg to 3.759 mg/kg at different solar drying conditions. The lowest TBARS value was obtained at a temperature of 30 °C, material thickness of 10 mm, and wind velocity of 0.9 m/s. The effects of three independent variables on TBARS values were obtained by the coefficient of the second-order polynomial regression equations in Table 4. Furthermore, neglecting nonsignificant terms, the fit model of TBARS values in coded levels was performed with Equation (7).

According to Equation (7), surface plots representing the relationship between process variables and TBARS values are presented in Figure 3B,C. The effects of temperature, material thickness, and wind velocity on TBARS values that had no significant difference are not shown in the present research (*p* > 0.05). Table 4 shows that the interaction effects between temperature and wind velocity and between the material thickness and wind velocity on TBARS values exhibited extremely significant differences (*p* < 0.05). Results showed that the interactive effects of temperature and wind velocity on TBARS values was significant (Figure 3B). Results further indicated that TBARS values decreased with increasing temperature and wind velocity. However, TBARS values increased as temperature increased when the wind velocity was high, which indicated that temperature could lead to the formation of secondary oxidation products in lipid oxidation [44]. Lipid oxidation may further play a significant role in color characteristic changes. This finding agreed with previous reports, which described that air temperature and air velocity affected the yellowness index of the nugget [45]. The interaction between material thickness and wind velocity on TBARS values was significant (Figure 3C). TBARS values initially decreased with increased material thickness and then increased. TBARS values decreased as wind velocity increased and at the maximum material thickness. This finding indicated that wind velocity could also affect the formation of secondary oxidation products. The reason might be because heat was removed with the flowing moisture by the wind velocity and decreased the degree of lipid oxidation. These results agree with those of previous reports [41]. As shown in Table 2, when the values of material thickness and wind velocity remained the same, TBARS values at high temperatures were smaller than those at low temperatures. These results are in agreement with the changes in the POV, which was remarkably affected by high temperatures due to the increased lipid oxidation reaction. Therefore, a high temperature can accelerate the decomposition of secondary oxidation products in lipid oxidation and can accelerate fat decomposition [41]. In the second stage of fat oxidation, TBARS values indicated fat oxidation and decomposition and showed undesirable flavor compounds [46]. 

### 3.4. Effects of Solar Drying Conditions on 5-HMF

The Maillard reaction is a common and complex chemical reaction in food processing and occurs in the carbonyl groups of reducing sugars and free amino groups of amino acids, peptides, and proteins. 5-HMF is a complex compound that forms as an intermediate product in the Maillard reaction or forms from the degradation of sugars under acidic conditions during heat treatment [21]. 5-HMF is also the main quality index employed to measure the Maillard reaction [16]. Temperature, sugar type [47], water activity [48], and drying time are the main factors of the increasing rate of the 5-HMF formation of dairy products [49]. In the present study, the 5-HMF contents changed from 5.38 mg/L to 23.67 mg/L at different solar drying conditions. In the experiments, the lowest 5-HMF content was obtained at a temperature of 46 °C, material thickness of 13 mm, and wind velocity of 1.2 m/s. The highest 5-HMF content was achieved at a temperature of 34 °C, material thickness of 7 mm, and the wind velocity of 0.6 m/s. The effects of drying temperature, material thickness, wind velocity, and their interactions on the formation of 5-HMF are presented in Table 3. Results clearly demonstrated that all the independent factors showed a significant effect on 5-HMF contents (*p* < 0.05) and that temperature had the strongest effect compared with the two other factors. The Maillard reaction can bring about a reduction in amino acids compounds and sugars at high temperatures. Moreover, the fit model of 5-HMF contents in coded levels was performed with Equation (8).

According to Equation (8), the surface plots in Figure 3D represent that the relationship between process variables and the interaction effect of temperature and material thickness was significant for 5-HMF contents (*p* < 0.05). The effects of temperature, material thickness, and wind velocity on 5-HMF contents that had no significant difference are not shown in the present research (*p* > 0.05). As shown in Table 2, 5-HMF contents were low at a temperature of 40–46 °C and the same material thickness and wind velocity. 5-HMF contents were low when the wind velocity was fast because the moisture content was removed rapidly. The Maillard reaction can be affected by many factors [50]. Moisture content played an important role in the generation of 5-HMF because the Maillard reaction rarely occurred when the reaction system had no moisture. However, the reaction rate will be fast when the moisture content is approximately 10–15% [2].

### 3.5. Optimization of Qula Solar Drying

The optimal conditions for Qula by solar drying were determined to obtain minimum POV, TBARS values, and 5-HMF contents. The second-order polynomial models obtained in the present study were used for each response for determining the specified optimum drying condition. These regression models were valid only in the selected experimental domain. Thus, the operating region was determined considering a few constraints related to economics, industries, and product quality. In the present study, the optimization was performed at a temperature range of 30–50 °C, material thickness of 5–15 mm, and wind velocity of 0.4–1.4 m/s. By applying the desirability function method, solutions were obtained at optimum drying conditions, i.e., temperature of 43.03 °C, material thickness of 10.72 mm, and wind velocity of 1.06 m/s. At this point, the POV, TBARS, and 5-HMF were 0.609 meq/kg, 0.504 mg/kg, and 4.553 mg/L, respectively.

### 3.6. Model Verification and the Comparison between Solar Drying and Open Sun-Drying

To confirm the accuracy and reliability of the predicted models and to evaluate the deviation between estimated and experimental values under the predicted optimal conditions, we performed a verification experiment on the basis of theoretical values. For convenient and precise manipulation, a minor modification was made to the optimal conditions. A verification experiment was performed at a temperature of 43.0 °C, material thickness of 11.0 mm, and wind velocity of 1.0 m/s, and under these modification conditions, the POV, TBARS, and 5-HMF of Qula were 0.65 meq/kg, 0.516 mg/kg, and 4.586 mg/L, respectively. No significant difference was observed between experimental and estimated values, thus verifying that the fit models for each response were valid and reliable. 

Figure 4 illustrates the products obtained through the optimized solar drying condition (A) and the open sun-drying method (B). Evidently, the quality of Qula subjected to solar drying was superior to that subjected to open sun-drying. The quality comparison of Qula produced by the two drying methods is summarized in Figure 5. As shown in Figure 5A–C, the POV, TBARS, and 5-HMF of Qula subjected to solar drying were significantly lower than those in the control group (*p* < 0.05), which indicated that the solar drying produced positive effects on the quality of Qula and was an efficient drying method for dairy products. However, the present results showed that the contents of moisture, protein, fat, and ash had no significant difference between the products in the two groups (*p* > 0.05, Figure 5D), which demonstrated that the present solar drying method had no influence on the nutritional qualities of Qula. 

The *L*^*^, *a*^*^, and *b*^*^ values of Qula were also measured and are important color evaluation indexes for dairy products. The *L*^*^ value is whiteness or brightness. A large *L*^*^ indicates a bright sample. The *a*^*^ value is redness or greenness. An increase in *a^*^* denotes a redder chroma, which is indicative of browning reaction. The *b*^*^ value is yellowness or blueness. A large *b*^*^ value indicates a yellower sample [51]. As shown in Figure 5E, the color evaluation of dried samples revealed that the surface color of solar-dried samples was different from that of open sun-drying samples. The *L*^*^ value of Qula subjected to solar drying was higher than that subjected to open sun-drying (*p* < 0.05). The *a*^*^ and *b*^*^ values represent red and brown colors, respectively, and small *a*^*^ and *b*^*^ values indicate improved color [52,53]. In the present study, the *a*^*^ and *b*^*^ values of the products obtained by solar drying were smaller than those of the products obtained open sun-drying (*p* < 0.05). The above differences were similar to the POV, TBARS, and 5-HMF of Qula. Our findings are consistent with the study that showed that lipid oxidation and the Maillard reaction are positively correlated with color [54]. This finding might be because, in the early drying process of Qula, large amounts of water are present, and different temperatures, which contribute to the fat oxidation and Maillard reaction, influence the color [20,55]. This finding agreed with the report that showed that the *a*^*^ and *b*^*^ values of dehydrated materials are strongly affected by temperature and air relative humidity. The effect of temperature on the changes of the *a*^*^ value during conventional drying seems to be more intense than the effect of air relative humidity for all materials [54]. Therefore, increasing the water evaporation rate can effectively decrease the quality deterioration. However, temperature and oxygen are prerequisites to the fat oxidation and Maillard reaction. The results of this study revealed that solar drying is an efficient method to prevent the quality degradation of Qula during drying.

### 3.7. Drying Characteristics

Based on the optimized solar drying conditions (temperature of 43.0 °C, material thickness of 11.0 mm, and wind velocity of 1.0 m/s), the drying curve was carried out (Figure 6). The drying curve of samples by open sun-drying was also performed, from 8 am to 6 pm each day (total of 26 h) and a temperature range of 25–35 °C during the effective drying process. The moisture content data observed in the drying experiment were converted into the moisture ratio (Figure 6). Solar drying remarkably enhanced the drying rate of Qula compared with open sun-drying, and a batch of drying can be completed within a day. During the initial stage, the removal of surface moisture caused the material to shrink fast and form a hard crust. Thus, the material became gradually impervious to moisture diffusion. Therefore, the surface becomes unsaturated, and the drying rate had a decreasing trend [45]. Our findings are consistent with the study that a solar dryer from solar energy systems is more promising than natural convection solar drying methods due to hygienic characteristics and short drying times [56]. 

The most suitable model can be found using three parameters, namely the correlation coefficient R^2^, root-mean-squared error (RMSE), and reduced sum-squared error *χ*^2^. The statistical results of seven different models, including the drying model coefficients and the comparison criteria used to evaluate the goodness of fit with R^2^, *χ*^2^, and RSME, are listed in Table 5. These drying models were employed to examine the drying kinetics of solar drying and open sun-drying of Qula. As shown in Table 5, in comparison with other models, the Midilli–Kucuk model could well describe the data of Qula during solar drying and open sun-drying (*R*^2^ > 0.99). Similarly, the Midilli–Kucuk model best simulated the lignite drying kinetics in all drying methods, including fluidized bed, vibrated fluidized bed, medium fluidized bed, and vibrated medium fluidized bed [24].

## 4. Conclusions

In the present study, Qula solar drying conditions were optimized by the RSM. The results revealed that the temperature of 43.0 °C, material thickness of 11.0 mm, and wind velocity of 1.0 m/s were the optimal conditions. Under these optimal conditions, the POV, TBARS, and 5-HMF were 0.65 meq/kg, 0.516 mg/kg, and 4.586 mg/L, respectively. Meanwhile, solar drying could effectively reduce the lipid oxidation and Maillard reaction of Qula compared with the open sun-drying method. The RSM was successfully applied to the parameter selection in the present study. The benefits of solar drying included shortened drying time and decreased oxidation and Maillard reaction. These features significantly improve the Qula quality, especially the color. The solar drying method and mixed-mode solar dryer can improve product quality and the use of the advantages of a rich solar energy resource. This method can be promoted and applied in the Qinghai-Tibet Plateau areas along with its remarkable potential application value for Qula products’ drying. 

## Figures and Tables

**Figure 1 foods-11-02406-f001:**
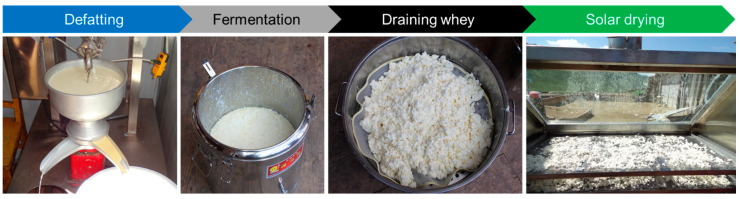
Material’s Preparation Scheme.

**Figure 2 foods-11-02406-f002:**
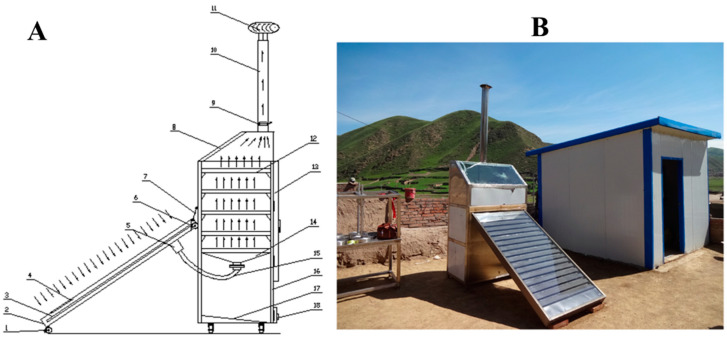
Schematic diagram (**A**) and picture of the facility (**B**) used for Qula solar drying. 1. Caster, 2. solar collector, 3. solar collector tube, 4. air inlet, 5. hot air collection filter, 6. pulley, 7. safety lanyard, 8. transparent window, 9. speed regulation board, 10. chimney, 11. self-rotating vacuum fan, 12. tray, 13. air-drying box door, 14. hot air distributor, 15. hot air connecting pipe, 16. air-drying box body, 17. air-drying box inclined bottom, 18. cleaning window.

**Figure 3 foods-11-02406-f003:**
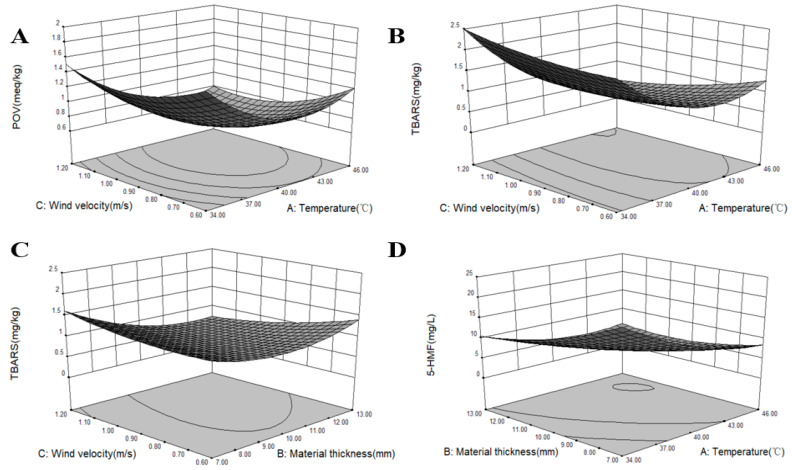
The effects of three optimization conditions on the POV, TBARS values, and 5-HMF contents. (**A**) The interactive effect of temperature and wind velocity on the POV. (**B**) The interactive effect of temperature and wind velocity on the TBARS values. (**C**) The interactive effect of wind velocity and material thickness on the TBARS values. (**D**) The interactive effect of temperature and material thickness on the 5-HMF contents.

**Figure 4 foods-11-02406-f004:**
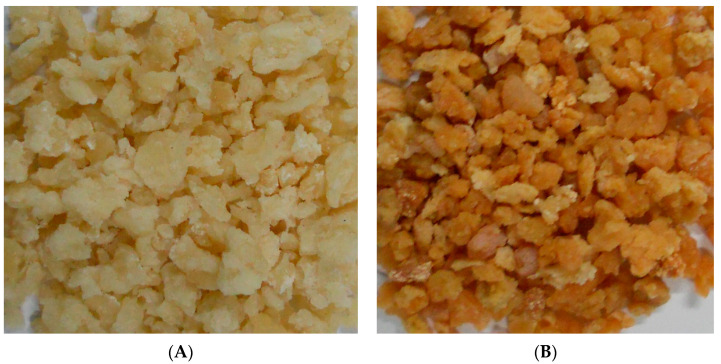
Comparison of Qula color from solar drying (**A**) and open sun-drying (**B**).

**Figure 5 foods-11-02406-f005:**
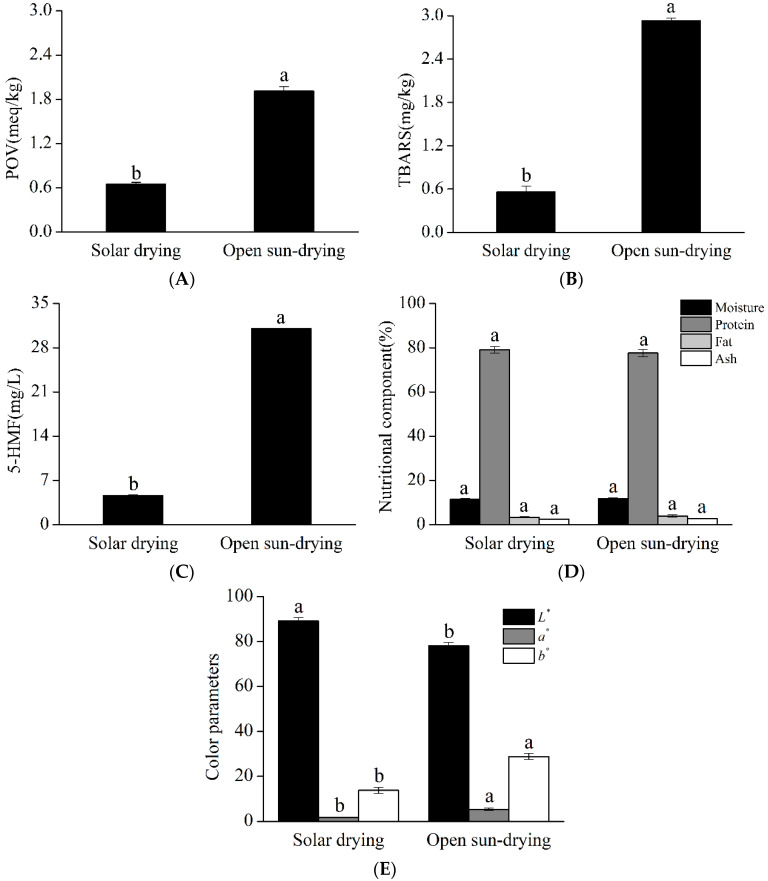
Quality comparison of Qula from solar drying and open sun-drying drying methods. Representative histograms showing the changes in the POV values (**A**), changes in the TBARS values (**B**), and changes in the 5-HMF contents (**C**); changes in the nutritional components (**D**) and color quality (**E**). ^a^^,b^ Means in the same color column with different superscripts differ significantly (*p* < 0.05).

**Figure 6 foods-11-02406-f006:**
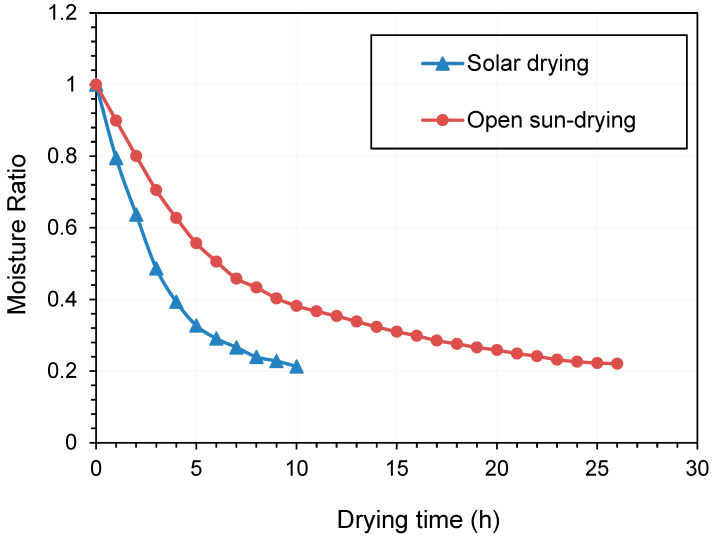
Drying kinetics of Qula by solar drying and open sun-drying.

**Table 1 foods-11-02406-t001:** Experimental values and coded levels of the independent variables in central composite rotatable design.

Independent Variables	Coded Levels				
	−1.68	−1	0	+1	+1.68
Temperature *X*_1_ (°C)	30	34	40	46	50
Material thickness *X*_2_ (mm)	5	7	10	13	15
Wind velocity *X*_3_ (m·s^−1^)	0.4	0.6	0.9	1.2	1.4

**Table 2 foods-11-02406-t002:** The CCD matrix and the experimental data for the responses.

Run	Factor Level Coding			Response		
	Temperature (°C)	Material Thickness (mm)	Wind Velocity (m/s)	POV (meq/kg)	TBARS (mg/kg)	5-HMF (mg/L)
	*X* _1_	*X* _2_	*X* _3_			
1	34 (−1)	7 (−1)	0.6 (−1)	2.15	2.367	23.67
2	46 (+1)	7 (−1)	0.6 (−1)	1.55	1.538	11.99
3	34 (−1)	13 (+1)	0.6 (−1)	1.88	2.546	13.15
4	46 (+1)	13 (+1)	0.6 (−1)	1.55	1.635	10.22
5	34 (−1)	7 (−1)	1.2 (+1)	2.06	3.360	18.81
6	46 (+1)	7 (−1)	1.2 (+1)	1.24	1.080	6.27
7	34 (−1)	13 (+1)	1.2 (+1)	1.77	2.258	11.28
8	46 (+1)	13 (+1)	1.2 (+1)	1.02	0.519	5.38
9	30 (−1.68)	10 (0)	0.9 (0)	2.08	3.759	22.47
10	50 (+1.68)	10 (0)	0.9 (0)	1.34	1.453	7.56
11	40 (0)	5 (−1.68)	0.9 (0)	1.78	2.077	20.02
12	40 (0)	15 (+1.68)	0.9 (0)	1.36	1.265	6.68
13	40 (0)	10 (0)	0.4 (−1.68)	1.45	1.293	16.56
14	40 (0)	10 (0)	1.4 (+1.68)	1.03	1.021	9.89
15	40 (0)	10 (0)	0.9 (0)	0.63	0.781	8.15
16	40 (0)	10 (0)	0.9 (0)	0.77	0.978	7.25
17	40 (0)	10 (0)	0.9 (0)	0.67	0.861	5.89
18	40 (0)	10 (0)	0.9 (0)	0.75	0.772	6.98
19	40 (0)	10 (0)	0.9 (0)	0.71	0.938	5.92
20	40 (0)	10 (0)	0.9 (0)	0.61	0.771	8.44

**Table 3 foods-11-02406-t003:** The drying models.

Models	Model Equations	References
Lewis	*MR* = exp(−*kt*)	Bruce (1985) [25]
Page	*MR* = exp(−*kt^n^*)	Page (1949) [26]
Henderson and Pabis	*MR* = *a*exp(−*kt^n^*)	Henderson and Pabis (1961) [27]
Logarithmic	*MR* = *a*exp(−*kt*) + *c*	Celma and Rojas (2007) [28]
Two term	*MR* = *a*exp(−*k*_0_*t*) + *b*exp(−*k*_1_*t*)	Henderson (1974) [29]
Midilli–Kucuk	*MR* = *a*exp(−*k*(*t^n^*)) + *bt*	Midilli and Kucuk (2003) [30]
Weibull	$MR=\exp\left[-{\left(\frac{t}{\beta }\right)}^{\alpha }\right]$	Wang and Yang (2018) [31]

**Table 4 foods-11-02406-t004:** Analysis of variance (ANOVA) of the fitted second-order polynomial model for POV, TBARS, and 5-HMF.

Source	*df*	POV			*df*	TBARS			*df*	5-HMF		
		Coefficient Estimate	Sum of Squares	*p*-Value		Coefficient Estimate	Sum of Squares	*p*-Value		Coefficient Estimate	Sum of Squares	*p*-Value
Model	7	0.69	5.14	<0.0001	9	0.85	15.18	<0.0001	9	7.16	629.97	<0.0001
*X* _1_	1	−0.27	1.03	<0.0001	1	−0.71	6.80	<0.0001	1	−4.26	247.39	<0.0001
*X* _2_	1	−0.11	0.16	0.0011	1	−0.20	0.55	<0.0001	1	−3.16	136.31	<0.0001
*X* _3_	1	−0.13	0.22	0.0003	1	−0.097	0.13	0.0079	1	−2.09	59.51	0.0004
*X*_1_X_2_					1	0.057	0.026	0.1654	1	1.92	29.61	0.0043
*X*_1_X_3_	1	−0.08	0.051	0.0344	1	−0.28	0.65	<0.0001	1	−0.48	1.83	0.3830
*X*_2_X_3_					1	−0.24	0.47	<0.0001	1	0.48	1.87	0.3782
*X* _1_ ^2^	1	0.38	2.09	<0.0001	1	0.63	5.72	<0.0001	1	2.44	85.99	<0.0001
*X* _2_ ^2^	1	0.33	1.58	<0.0001	1	0.30	1.29	<0.0001	1	1.85	49.54	0.0008
*X* _3_ ^2^	1	0.21	0.66	<0.0001	1	0.12	0.20	0.0021	1	1.81	47.20	0.0009
Residual	12		0.11		10		0.12		10		22.02	
Lack of fit	7		0.087	0.1227	5		0.076	0.2588	5		16.23	0.1413
Pure error	5		0.021		5		0.041		5		5.79	
Total	19		5.25		19		15.30		19		651.99	
*R* ^2^			0.9794				0.9923				0.9662	
Adj-*R*^2^			0.9674				0.9854				0.9358	
Pre-*R*^2^			0.9210				0.9533				0.7974	
Adq-precision			25.645				44.283				18.112	
*PRESS*			0.41				0.71				132.06	
*C.V.*			7.19				6.94				13.10	

**Table 5 foods-11-02406-t005:** Statistical results obtained from drying models in solar drying and open sun-drying.

Models	Solar Drying				Open Sun-Drying			
	Estimated Parameters	*χ*^2^ (×10^−3^)	*RSS*	*R* ^2^	Estimated Parameters	*χ*^2^ (×10^−3^)	*RSS*	*R* ^2^
Lewis	*k* = 0.2030	2.2000	0.0220	0.9678	*k* = 0.0825	4.8100	0.1253	0.8971
Page	*k* = 0.2827 *n* = 0.7929	0.9070	0.0082	0.9867	*k* = 0.1880 *n* = 0.6716	0.8115	0.0203	0.9827
Henderson and Pabis	*a* = 1.0140 *k* = 0.2934 *n* = 0.7793	0.9934	0.0080	0.9855	*a* = 1.0393 *k* = 0.2123 *n* = 0.6395	0.7642	0.0183	0.9837
Logarithmic	*a* = 0.8378 *k* = 0.3215 *c* = 0.1743	0.1694	0.0014	0.9975	*a* = 0.6274 *k* = 0.1603 *c* = 0.2242	0.1554	0.0037	0.9967
Two term	*a* = 0.9544 *k*_0_ = 0.2712 *b* = 0.0529 *k*_1_ = 0.1086	0.1265	8.8522 × 10^−4^	0.9982	*a* = 0.6274 *k*_0_ = 0.2128 *b* = 0.3898 *k*_1_ = 0.0225	0.0733	0.0017	0.9984
Midilli–Kucuk	*a* = 1.0012 *k* = 0.2521 *n* = 1.0620 *b* = 0.0168	0.8575	6.0024 × 10^−4^	0.9988	*a* = 1.0186 *k* = 0.1595 *n* = 0.8497 *b* = 0.0059	0.2245	0.0052	0.9952
Weibull	*β* = 4.9208 *α* = 0.7925	0.9070	0.0082	0.9867	*β* = 12.0424 *α* = 0.6712	0.8115	0.0203	0.9827

## Data Availability

The data presented in this study are available on request from the corresponding author.

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
