# Peer review of "Response Surface Optimization of Solar Drying Conditions and the Effect on the Quality Attributes and Drying Characteristics of Qula Casein"

_foods, 2022, doi:10.3390/foods11162406_

Round 1

Reviewer 1 Report

The manuscript presents the effects of solar drying on quality attributes and drying characteristics of Qula casein. The work is interesting from the point of view of characteristics of Qula casein. In my opinion, the work can be improved by attending the follow minor revisions.

1 Hue, chroma, total color difference and browning index must be included in the manuscript.

2.      Line 353. b value represents yellow color.

3.      Colorimetric analysis should be improved by including hue angle, chroma or saturation and browning index

4.      Final moisture content and water activity are important parameters which are correlated with the shelf life of dried products; therefore, final moisture content and water activity in the final product must be reported and expressed in a surface methodology plot. You can revise and reference the following article in your introduction

     https://doi.org/10.24275/rmiq/Alim2652

Izadi, Z., Mohebbi, M., Shahidi, F., Varidi, M. and Reza, S.M. (2020). Cheese powder production and characterization: A foam-mat drying approach. Food Bioproducts Process, 123, 225–237. https://doi.org/10.1016/j.fbp.2020.06.019.

McCarthy, C.M., Wilkinson, M.G., Kelly, P.M. and Guinee, T.P. (2015). Effect of salt and fat reduction on the composition, lactose metabolism, water activity and microbiology of cheddar cheese. Dairy Science and Technology 95, 587-611. https://doi.org/10.1007/s13594-015-0245-2

Soares, C., Fernando, A.L., Mendes, B. and Martins, A.P.L. (2015). The effect of lowering salt on the physicochemical, microbiological and sensory properties of Sao Joao cheese of Pico Island. International Journal of Dairy Technology 68, 409-419 DOI: 10.1111/1471-0307.12198 

Optimization of spray drying process in cheese powder production

https://doi.org/10.1016/j.fbp.2013.12.008

6.      Water activity is related with the lipid oxidation, therefore the final moisture content and water activity must be included in Table 2 for the 20 experiments

7.      Physicochemical, and microbiological properties of raw Qula casein must be reported.

8.      Physicochemical and microbiological properties of dried Qula casein must be reported.

9.      How the authors can ensure the stability of the dried product?

1  How the authors can control the air velocity (0.4, 0.6, 0.9, 1.2, and 1.4 m/s) in the solar dryer?

Reviewer 2 Report

The manuscript entitled “Effects of solar drying on quality attributes and drying characteristics of Qula casein” is awkwardly written.

The reviewer comments are as the followings:

In section 2.4.4, the nutritional components are reported to be measured, however not reported in the section R&D.

The results are reported, however, without any comparisons to the previous studies with sufficient discussion.

Only the between group multiple range comparisons are conducted, there are no within-group multiple range comparisons for the solar drying and open sun-drying     

English language is always problematic throughout the entire manuscript.

Some of the English problems are listed but not limited to these mentioned ones:

Line 17 ?? In these conditions (there is only one optimum condition).

Line 29 ?? It is not fundamental of… (not clear)

Line 40 ?? issue that raw material itself. (non-readable)

Line 54 ?? at the range (non-readable)

Line 56 ?? its’ (non-readable)

Line 57 ?? need to be clear (non-readable)

Line 64 ?? Maillard’s reaction (non-readable)

Line 73 ?? The yak milk obtained from (non-readable)

Line 79 ?? the method of [2]. (incomplete sentence)

There might as well be some consistency problems such as hot air velocity (in abstract) versus wind velocity (in M&M).

The authors should value and respect the time and efforts of editors and reviewers during the preparatory written process.  

Reviewer 3 Report

there are many comments in the manuscript must be revised according to it. there is a confuse in determine temperature and wind velocity at deferent levels.

Round 2

Reviewer 1 Report

the authors answered appropriately to the comments made on the previous version

Reviewer 2 Report

The manuscript has been significantly improved!